# Fruiting Characteristics and Molecular-Assisted Identification of Korla Fragrant Pear Bud Mutation Materials

Xian'an Yang [1,2], Cuifang Zhang [1,2], Haichang Sun [3], Shiwei Wang [1,2,*], Yutong Cui [1,2] and Long Zhao [1,2]

1 College of Forestry and Landscape Architecture, Xinjiang Agricultural University, Urumqi 830052, China; 18095950962@163.com (X.Y.); zcf851022@163.com (C.Z.); 18040796669@163.com (Y.C.); 13161097837@163.com (L.Z.)
2 Key Laboratory of Forestry Ecology and Industry Technology in Arid Region, Xinjiang Agricultural University, Urumqi 830052, China
3 Luntai County Apricot Research and Development Center, Luntai 841600, China; sunh7758521@126.com
* Correspondence: wsw850204@163.com

**Abstract:** Korla fragrant pear is a high-quality local pear variety native to Xinjiang, China. Currently, the low fruit-setting rate and low calyx shedding rate problems in Korla fragrant pears have been highlighted, which seriously affect the fruit yield and quality. It is of great significance to research the fruiting characteristics and molecular-assisted identification of Korla fragrant pear bud mutation materials for enriching the germplasm resources of Korla fragrant pear. In this research, a natural pollination group (YB) of Korla fragrant pear bud mutation materials and a natural pollination group (CK) of Korla fragrant pears were established. On the fruiting characteristics, the fruit-setting rate and calyx-removal rate of the two groups were investigated. In terms of fruit quality, the fruit shape index, fruit specific gravity, soluble solids content, sugar:acid ratio, soluble sugar content, and other indicators were measured. For the anatomical structure of the calyx tube, the detachment cells were observed. The formation time of the two groups of detached cells was compared. In the GBS simplified genome sequencing, a phylogenetic tree was constructed based on the obtained SNP sites. A principal component analysis, population genetic structure analysis, and genetic diversity index analysis were carried out. In the aspect of SSR molecular marker identification, the SSR types were counted. Polyacrylamide gel electrophoresis was performed. The results demonstrate the following: (1) the fruit-setting rate (30.87%) and calyx-removal rate (68.11%) in the YB group were significantly higher than those in the CK group (19.37%) and the calyx-removal rate (55.18%). (2) There was no significant difference in fruit quality indexes, such as average fruit weight (127.10–130.00 g) and soluble sugar content (9.47–9.56%) between the two groups. (3) Abscission-layer cells were observed at 2, 4, 6, 8, and 10 h after calyx tube discoloration in the YB group and at 48, 72, and 96 h after calyx tube discoloration in the CK group. (4) The genetic background of the YB group and the CK group was similar at the GBS level, but there were differences at the DNA level. This research finally shows that Korla fragrant pear bud mutation material is a good germplasm resource. This germplasm resource can promote the structural optimization of Korla fragrant pear varieties and the healthy development of the industry.

**Keywords:** Korla fragrant pear; bud mutation; fruiting characteristics; delaminated cells; molecular identification





## 1. Introduction

Korla fragrant pear (*Pyrus sinkiangensis* Yu) is a perennial deciduous fruit tree of the genus Pyrus in the family Rosasceae, and it is a high-quality local pear variety that is native to Xinjiang, China. Additionally, it is a geographical indicator product for China. Because of its unique aroma, thin and smooth peel, delicate and juicy pulp, crisp and refreshing taste, and excellent quality, it is known as the 'best pear' locally and abroad [1,2]. Korla fragrant pear is mainly planted in China's Xinjiang Bayinguoleng Mongolian Autonomous

Prefecture and in a few counties and cities in the Aksu region [3]. In recent years, the export market of the Korla fragrant pear has expanded from Southeast Asia to many countries, such as the United States, Canada, and Australia, and has become the main fruit for China's Xinjiang export earnings [4,5]. Currently, the Korla fragrant pear has a low fruit-setting rate under natural conditions and numerous persistent calyx fruits. The persistence of the calyx affects the shape and quality of Korla fragrant pear fruits, affects the high-quality development of the Korla fragrant pear industry, and causes serious economic losses [6–8]. To solve this problem, fruit growers usually use artificial powdering to improve the fruit-setting rate during the cultivation and management of Korla fragrant pears, and spray plant growth regulators during the flowering period to improve the calyx-removal rate. Studies have shown that the fruit-setting rate of Korla fragrant pear free-pollinated pear trees is significantly lower than that of artificially pollinated pear trees [3]. Spraying IAA at the flowering stage of Korla fragrant pear can promote calyx persistence, whereas spraying TIBA can significantly increase the calyx-removal rate of Korla fragrant pear [9]. When PP333 was sprayed at the flowering stage of Korla fragrant pears, there were only vessels in the calyx tubes of young fruits, and there were no sieve tube cells or heterocytes. At the late calyx tube development stage, an abscission layer appears, the calyx tube falls off, and a calyx-free fruit finally forms [10]. Although artificial pollination can improve the fruit-setting rate of Korla fragrant pear, it is very costly for large-scale planting orchards [3]. Plant growth regulators can increase the calyx-removal rate of Korla fragrant pears, but excessive plant growth regulator use affects fruit quality. Studies have shown that spraying NAA plant growth regulators can promote fruit thinning, but simultaneously cause plant leaf deformity, inhibit fruit hypertrophy, and lead to dwarf fruit [11]. Spraying uniconazole can effectively increase the calyx-removal rate and fruit weight of Korla fragrant pears, but inhibits the growth of new shoots, internode length, and leaf area of Korla fragrant pears, increases the coarse-skinned fruit ratio, and increases fruit hardness and stone cell content [12]. Nowadays, consumers pay more attention to the quality and safety of agricultural products, and the demand for green organic agricultural products is increasing. Therefore, to promote the healthy and sustainable development of the Korla fragrant pear industry and meet the growing demand for green organic food by consumers today [13–15], breeding varieties with excellent fruiting characteristics under natural conditions is the most effective measure.

Korla fragrant pears are mostly bred using bud mutations. Compared to traditional hybrid breeding methods, bud mutation breeding [16–18] can avoid interference in early development, has a short periodicity, and can improve individual shortcomings while essentially retaining the comprehensive traits of the original variety, thereby streamlining the breeding process and shortening the breeding cycle. Currently, the bud mutation varieties of Korla fragrant pear mainly include the 'Sha 01' line and the 'Xinli 2' line [19,20]. Compared with Korla fragrant pear, the bud mutation characteristics of the 'Sha 01' and 'Xinli 2' strains were mainly reflected in the large single fruit volume, low stone cell content, and early maturity, but the low fruit-setting rate and low calyx-removal rate problems were not improved. In terms of fruit quality, the unique flavor of Korla fragrant pears is composed of several quality indicators, but the existing bud mutant varieties (lines) failed to improve one or more quality indicators while maintaining the original quality. In this research, we compared the fruit quality of Korla fragrant pear bud mutant material with that of Korla fragrant pear to better understand the fruit quality of Korla fragrant pear bud mutant material under natural conditions.

The breeding of bud mutation materials plays an important role in fruit tree breeding [21]. However, whether the bud mutation material can be used and whether it is a high-quality germplasm resource need to be identified. At present, GBS sequencing technology and molecular marker technology have been widely used in the identification of plant germplasm resources. GBS sequencing technology can be used to analyze the genetic background and genetic diversity of different germplasm resources [22,23]. Analysis of population genetic diversity is crucial for cultivating new varieties and evaluating and

utilizing plant germplasm resources. They are an important component of plant genetics, breeding, protection, and evolution [24]. This has been reported in the research of germplasm resources of maize (*Zea mays*) [25], rice (*Oryza sativa*) [26], pepper (*Capsicum* spp.) [27], *Camelina sativa* [28], watermelon (*Citrullus lanatus*) [29], and other species. It can be seen that GBS sequencing technology is of great significance in the identification, preservation, and utilization of germplasm resources. In addition, SNP loci obtained by GBS sequencing technology can be used for genome-wide association analysis (GWAS). For example, in the research of 95 *Brassica napus* [30] germplasm resources, it was determined that based on the SNP loci obtained by GBS sequencing, the GWAS method was the best to determine the genes related to 1000-grain weight (*SFGH*, *ZBED1*, *VLN3*, *DLC1*). There was also research that combined GBS sequencing and GWAS methods to identify genes (*MYB16-like*, *R2R3-MYB*) related to the prickle development of red raspberry (*Rubus idaeus*) [31]. This indicates that GBS sequencing can provide useful information for GWAS analysis. The combined application of GBS and GWAS plays an important role in finding species trait genes. Molecular marker technology is a common means of plant genetic breeding research [32]. Among different types of molecular markers, SSR molecular marker technology is widely used. This has been reported in the identification of 101 peanut (*Arachis hypogaea*) [33] varieties, 149 melon (*Cucumis melo*) [34] varieties, 11 walnut (*Juglans regia*) [35] varieties, and other species. This indicates that SSR molecular marker technology can be used for germplasm resources identification. Although there has been a lot of research conducted on the species resources of different plants at this stage, there are few research studies on the genetic diversity of Korla fragrant pear and bud mutation materials. The analysis and identification of bud mutation materials is very important for the utilization of this germplasm resource. Therefore, GBS simplified genome sequencing and SSR molecular marker identification were performed in this research to better understand the genetic differences between Korla fragrant pear and its mutants under natural conditions.

In summary, to further clarify the differences in the seed-setting characteristics and genetic differences between the two, this research used Korla fragrant pear bud mutation material and Korla fragrant pear as the research objects, through the investigation of seed-setting characteristics, anatomical structure observations, molecular markers, and other technical means, aiming to answer three main questions:

(1) Is there any difference in the fruiting characteristics between Korla fragrant pear bud mutant material and Korla fragrant pears?
(2) Are there any differences in the structural characteristics of sepal (cylindrical) cells between Korla fragrant pear bud mutation materials and Korla fragrant pears?
(3) Does the degree of variation between Korla fragrant pear bud mutation materials and Korla fragrant pears affect population evolution?

## 2. Materials and Methods

### 2.1. Experimental Site and Plant Materials

This research was conducted from March to September 2022. The test site is located in Luntai County, Bayinguoleng Mongolian Autonomous Prefecture, Xinjiang, China. The geographical coordinates are 41°06′–42°32′ N, 83°38′–85°26′ E, with an average altitude of 1060 m. Luntai County belongs to the temperate continental arid climate. The annual average sunshine hours are approximately 2783 h, the frost-free period is approximately 188 d, the annual average temperature is 10.6 °C, the annual average precipitation is 52 mm, and the annual potential evaporation is 2072 mm. The test materials were the grafting tree of the Korla fragrant pear bud mutation material and the Korla fragrant pear tree (Figure 1). The grafted tree age of the Korla fragrant pear bud mutation material was 10 years, the Korla fragrant pear tree age was 10 years, and the row spacing of the plant was 4 × 5 m. The Dangshan pear was used as the pollination tree in the garden, and the configuration ratio of the main varieties to the pollination tree varieties was 10:1. The cultivation and management conditions in the garden were consistent, and the trees were healthy. We

randomly selected 10 grafted trees of Korla fragrant pear bud mutation materials with the same management measures and site conditions, which had basically the same growth and no pests and diseases as the natural pollination group (YB) of the Korla fragrant pear bud mutation materials and selected 10 natural pollination trees of Korla fragrant pear as the natural pollination group (CK) of the Korla fragrant pear. Two test groups were established in this research. In the four directions, east, west, south, and north of each tree crown, two branches with the same growth potential, length, and thickness were selected as reference sample branches. Each tree was marked with eight branches, and each test group was marked with 80 branches to investigate the flower fruit-setting rate. Five trees were randomly selected from each group, and the marked fruiting branches were used to investigate the calyx-removal rate. After ripening (mid-September), one fruit was randomly collected from each of the 10 trees in each group in four directions: east, west, south, and north. A total of 40 fruits were collected from each test group to determine the fruit quality.

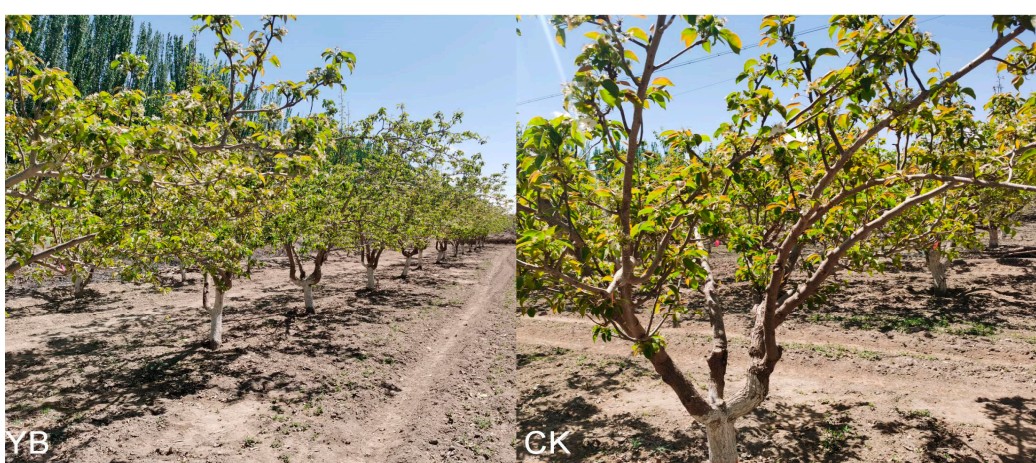

**Figure 1.** The distribution map of the trees in the pear orchard. YB represents the grafting tree of Korla fragrant pear bud mutation material, and CK represents the Korla fragrant pear tree.

*2.2. Investigation of Fruiting Characteristics*

(1) Fruit-setting rate: the number of fruit-bearing flowers accounted for the percentage of total flowers. The number of flowers on all the branches in each test group was counted during the flowering period. When the fruit reached the fruit-setting standard (diameter of 0.8 cm), the number of flowers on all the branches was counted, and the fruit-setting rate of each branch was calculated. The average fruit-setting rate of all the branches was used as the fruit-setting rate of the flowers in the test group.

$$\text{Fruit-setting rate of each marked branch (\%)} = \frac{\text{the number fruit-bearing flowers}}{\text{total flowers}} \quad (1)$$

$$\text{Fruit-setting rate (\%)} = \frac{\text{sum of fruit-setting rates of all labeled branches}}{\text{number of all tagged branches}} \quad (2)$$

(2) Calyx abscission rate: the percentage of fruits with calyx abscission to the total fruit number. Natural fruit drop occurs on the fragrant Korla pear tree branches. When the color of the young fruit calyx tube begins to change slowly, it is considered a critical period of shedding. Starting from the critical period of shedding, the number of fruit settings and calyx fruits on all the marked branches were investigated daily until the number of calyx fruits no longer changed. The average calyx-removal rate of all the branches was used as the calyx-removal rate for the test group.

$$\text{Calyx abscission rate of each marked branch (\%)} = \frac{\text{the number of fruits with calyx abscission}}{\text{total fruit number}} \quad (3)$$

$$\text{Calyx abscission rate } (\%) = \frac{\text{sum of calyx abscission rates of all labeled branches}}{\text{number of all tagged branches}} \qquad (4)$$

(3) Fruit quality: in terms of appearance quality measurement, 10 fruits were randomly selected from the fruits collected in each test group. A Vernier caliper was used to measure the transverse diameter (cm) and longitudinal diameter (cm) of the fruit, and the fruit shape index (the longitudinal diameter to transverse diameter ratio) was calculated. The single fruit weight (g) was measured with an electronic balance. The volume ($cm^3$) of the fruit was measured by the drainage method, and the specific gravity ($g/cm^3$) of the fruit (the weight to volume ratio) was calculated. After removing the core, the pulp was weighed (g) and the edible rate (%) (the ratio of pulp weight to single fruit weight) was determined. Ten fruits were randomly selected from the fruits of each test group, and the fruit skin hardness ($kg \cdot cm^{-2}$) and fruit peel hardness ($kg \cdot cm^{-2}$) were measured using a GY-B3 fruit hardness tester. In terms of the internal quality determination, 10 fruits were randomly selected from the fruits of each test group, and the soluble solid content (%) in the fruits was determined by the refractometer method [36]. The total acid content (g/kg) in the fruit was determined by the determination method of total acid in food, and the ratio of sugar to acid (the ratio of soluble solids content to total acid content) was calculated [37]. The soluble sugar content (%) of fruit was determined by the 3,5-dinitrosalicylic acid colorimetric method [38], and the fruit moisture content (%) was determined by the moisture content determination method in food [39]. The vitamin C content (mg/100 g) in the fruit was determined by the ascorbic acid determination method in food [40]. The soluble protein content (mg/g) of the fruit was determined by Coomassie brilliant blue staining [41]. The stone cell content (%) (the stone cell weight to pulp weight ratio) was determined by the gravimetric method [42].

### 2.3. Observation of the Anatomical Structure of the Calyx Tube

There are two cases of calyx shedding and persistence during fragrant Korla pear growth and development. Calyx shedding can be divided into two types: sepal shedding and calyx tube shedding (Figure 2). Field observations revealed that the fruits of the two calyx-shedding methods could grow and develop into calyx-shedding fruits. However, the fruit shape of calyx shedding was slightly different from that of calyx tube shedding.

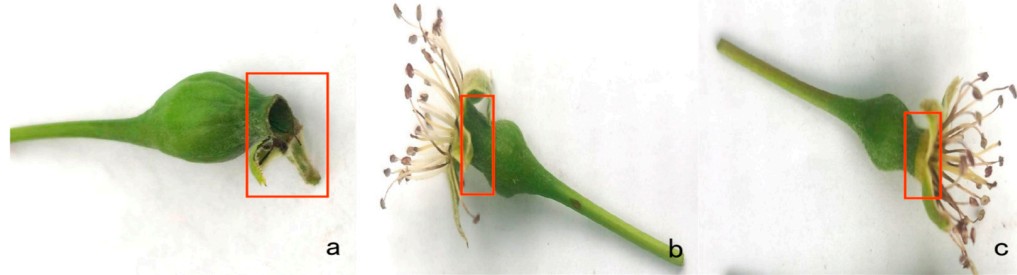

**Figure 2.** Diagram of calyx shedding. The sepal falls off from (**a**), (**b**) is the calyx reservoir, and (**c**) is the calyx cylinder falling off (coming off). The red box indicates the connection between the calyx tube or sepal and the young fruit.

The cell morphological parts of the sepals and calyx tube were identified based on the calyx tube anatomy in Korla fragrant pears. In the corresponding position of the sepal and the calyx tube (Figure 3), the upper bending part is the connection between the sepal and the calyx tube, and the lower end is the calyx tube area and the connection between the calyx tube and young fruit.

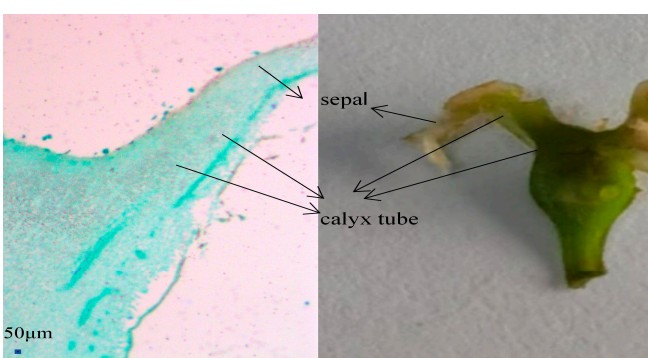

**Figure 3.** Korla fragrant pear young fruit 2.5× objective lens picture and the corresponding tissue structure picture.

Sepal (cylindrical) discoloration occurred first in the YB group. Therefore, based on the sepal (cylinder) discoloration in the YB group, sampling was performed at 2, 4, 6, 8, 10, 12, 24, 36, 48, 72, 96, 120, 144, and 168 h after discoloration, with 2, 4, 6, 8, and 10 h as the first period, 12, 24, and 36 h as the second period, 48, 72, and 96 h as the third period, and 120, 144, and 168 h as the fourth period. The sampling site was the tissue at the junction of the calyx tube and young fruit. Five samples were collected at each time point and placed in an FAA fixative (70% alcohol, 90 mL; glacial acetic acid, 5 mL; formaldehyde, 5 mL; and glycerin, 5 mL) for paraffin sectioning. After the sectioning was completed, the safranin-fast green double-staining method was used for staining, and the neutral gum was sealed and dried in an electrothermal constant temperature blast oven at 40 °C [43–46]. The prepared seals were observed under an optical microscope (Leica DM6000B, Germany) and photographed using a microscope camera, and the images were observed and compared.

*2.4. Genotyping by Sequencing*

Collection of test materials: A total of 10 trees were selected from each of the YB and CK groups, and 10 tender leaves were collected from each tree as samples. The samples were numbered YB1–YB10 and CK1–CK10. The leaves were placed in a sealed bag and stored with a silica gel desiccant. DNA extraction was performed using a DNA kit produced by Kangwei Century Biological Company (Beijing, China) according to the instructions. Construction and library inspection of GBS gene library [47,48]: The extracted DNA samples were commissioned by Novo Bioinformatics Technology Co., Ltd. (Beijing, China) for sequencing and database construction. First, one or more of the enzymes Mse I, Nla III, Hae II, Hae III, Msp I, and EcoR I were used for digestion evaluation. After evaluation, the first DNA digestion was performed, and Mse I enzyme was also used. Solexa P1 and P2 adapters were added at both ends of the digested fragment (which could recognize the Mse I restriction enzyme site and complement the enzyme digestion end). Then, the second enzyme selected in advance was used for combined enzyme digestion (adjusting the number of tags). The sequences containing P1 and P2 adaptors at both ends were amplified by PCR, and the DNA bands were recovered by electrophoresis. The PCR products were purified by AMPure XP beads to obtain the GBS library. After the construction of the library was completed, the preliminary quantification was performed using a Qubit2.0 Fluorometer, and the library was diluted to 1.5 ng/uL. Subsequently, Agilent 2100 bioanalyzer (Beijing Longyue Biotechnology Development Co., Ltd., Beijing, China) was used to detect the insert size of the library. After the insert size was in line with expectations, the effective concentration of the library was accurately quantified by qRT-PCR to ensure the quality of the library. On-machine sequencing: After the library inspection was qualified, different libraries were pooled according to the effective concentration and the target offline data volume requirements, and the sequencing was completed on the Illumina platform, thereby generating 150 bp paired-end readings.

SNP detection: The original data were filtered to filter out the reads containing the adaptor sequence. When the N content in the single-end sequencing read exceeded 10% of

the read length ratio, the paired reads were removed. When the number of low-quality bases (quality value q ≤ 5) contained in the single-end sequencing read exceeded 50% of the read length ratio, this pair of paired reads was removed. After strict filtering of the sequencing data, high-quality clean data were finally obtained. Comparison of reference genome: Chinese white pear (https://www.ncbi.nlm.nih.gov/genome/?Term=Pyrus+bretschneideri. Accessed on 7 October 2022) was used as the reference genome. Through the comparison software BWA (version 0.7.17) [49], the effective high-quality sequences filtered from each sample were compared with the reference genome, and the parameter was set to mem-t4-k32-M. At the same time, SAMTOOLS (version 1.17) [50] software was used to detect the SNP of 20 samples. After filtering the SNP sites detected by SAMTOOLS software, high-quality SNP sites were finally obtained for subsequent analysis. Phylogenetic tree analysis: In order to evaluate the phylogenetic evolution of 20 pear resources, we used TreeBest (http://treesoft.sourceforge.net/treebest.shtml. Accessed on 26 October 2022) software to calculate the distance matrix based on SNP genotyping results. In addition, we used the obtained genetic matrix to construct a phylogenetic tree by adjacency method. Among them, the initial value of the bootstrap value was set to 1000. Principal component analysis: According to the results of the SNP genotyping, GCTA software (Version 1.5) was used to calculate the eigenvalues and eigenvectors, and R software (Version 4.0.2) was used to draw the PCA result map. Population genetic structure analysis: We used PLINK (https://zzz.bwh.harvard.edu/plink/ Accessed on 12 November 2022.) to convert the obtained SNP vcf file into a bed file format. Meanwhile, admixture (http://dalexander.github.io/admixture/ Accessed on 27 November 2022) software was selected to determine the optimal K value based on the cross-validation method. We extracted the cross-validation error rate when K = 2–6, and the cross-validation error rate value gradually increased. Finally, the minimum error rate K = 2 was selected as the best K value for mapping. Analysis of population heterozygosity and genetic diversity index: Based on the filtered SNP loci of 20 samples, we used Arlequin (Version 3.5.2.2) software to calculate the observed heterozygosity (Ho), expected heterozygosity (He), and genetic diversity index.

### 2.5. SSR Molecular Marker Primer Screening

The sequencing method was the same as that used in the GBS simplified genome-sequencing process. Genome assembly and reference genome SSR development: Assembly was accomplished using SOAPdenovo [51] (SOAPdenovo-63mer: Version 2.04) with the parameter set to 'SOAPdenovo-63mer all-K 47.' Using the script perl to detect simple repetitive sequences in DNA sequences, primer 3 [52] was used for primer design. The DNA was extracted using the DNA kit produced by Kangwei Century Biological Company (Beijing, China). In order to detect the quality of the DNA, the absorbance values (OD) at 260 nm and 280 nm were measured by ultramicro ultraviolet spectrophotometer (nanodrop one). PCR amplification: The required primers were synthesized by Shanghai Sangon Biotech Co., Ltd. (Shanghai, China). The PCR reaction system was 25 μL, and the primer was developed by Chen [53]. The specific procedure was pre-denaturation at 95 °C for 3 min; denaturation at 95 °C for 30 s; annealing at 55 °C for 1 min; extension at 72 °C for 30 s, 32 cycles; extension at 72 °C for 10 min; and holding at 4 °C for 10 min. Polyacrylamide gel electrophoresis [54–56]: In this experiment, a vertical electrophoresis tank was used to measure 70 mL of 6% polyacrylamide glue, and 30 μL tetramethylethylenediamine (TEMED) and 300 μL ammonium persulfate (APS) were then added. After stirring with a glass rod, the glue was poured. When the glue formed a straight line at the other end, the glass plate was flattened, and a shark ruler was inserted on the toothless side, waiting for the glue to solidify. Electrophoresis: 1 × TBE buffer was added to the electrophoresis tank, the gel surface was cleaned, the side with teeth was inserted, and the sample was pointed into the gel surface. After the sample was spotted, the voltage was adjusted to 150 V, and electrophoresis was started. Silver staining: After electrophoresis, the plate was placed in a plastic box and rinsed with distilled water. Distilled water (1.5 L) and 1.5 g

AgNO3 were added to the dyeing box, and silver staining was performed for approximately 8 min. After washing the dyed plate in distilled water, the plate was placed in a configured developer solution (24 g NaOH and 12 mL CH2O in distilled water) for development. After development, it was rinsed with distilled water and dried to obtain the photographs.

### 2.6. Statistical Analysis

In this research, at least 3 biological replicates were set for each experiment. The standard deviation (SD) is expressed by the length of the error bar. We used *t*-test to analyze the experimental data. The criteria of statistical difference were $p < 0.05$ or $p < 0.01$. Excel (version 2016) was used for data collection, and IBM SPSS Statistics 26 software was used for data analysis. Plotting was performed using GraphPad Prism 8; the distance matrix was calculated using TreeBest software (version: 1.9.2). A phylogenetic tree was constructed using the adjacency method. A PCA distribution map was drawn using R4.0.2. The group structure was analyzed using Linux 1.3.0 software.

## 3. Results

### 3.1. Comparison of Fruiting Characteristics of Korla Fragrant Pear and Its Bud Mutation Materials

The fruiting characteristics of the Korla fragrant pear and its bud mutation materials were compared, as shown in Figure 4. Under natural conditions, the fruit-setting and calyx-shedding rates of flowers in the YB group were significantly higher than those in the CK group ($p < 0.01$) (Figure 4A). In terms of fruit appearance, there was no significant difference between the YB group and CK group in the average single fruit weight, edible rate, fruit specific gravity, fruit hardness, or fruit shape index ($p > 0.05$) (Figure 4B,C). The average single fruit weight was 127.10–130.00 g, the edible rate was 75.19–76.72%, the fruit specific gravity was 0.7838–0.7843 g·cm$^{-3}$, the fruit skin hardness was 7.15–7.16 kg·cm$^{-2}$, the peel hardness was 4.04–4.20 kg·cm$^{-2}$, and the fruit shape index was 1.04–1.05. In terms of internal fruit quality, there were no significant differences between the YB group and CK group in terms of soluble solid content, soluble sugar content, total acid content, sugar-acid ratio, stone cell content, fruit water content, ascorbic acid content, and soluble protein content ($p > 0.05$) (Figure 4D,E). The soluble solids content was 10.56–10.59%, the soluble sugar content was 9.47–9.56%, the total acid content was 0.60–0.61 g/kg, the sugar:acid ratio was 17.47–17.68, the stone cell content was 0.305–0.310%, the fruit water content was 85.49–87.64%, the ascorbic acid content was 3.77–3.78 mg·100 g$^{-1}$, and the soluble protein content was 0.24–0.26 mg·100 g$^{-1}$. The results above demonstrate that the bud mutation materials of Korla fragrant pear were consistent with the fruit quality of Korla fragrant pear, retaining the original flavor of Korla fragrant pear. However, the fruit-setting rate and calyx-removal rate were significantly improved.

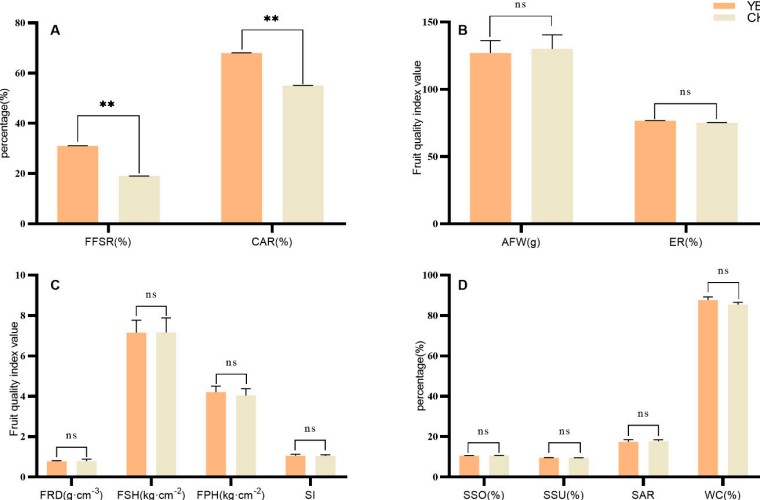

**Figure 4.** *Cont.*

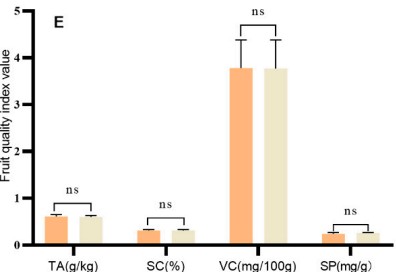

**Figure 4.** Comparison of fruiting characteristics of Korla fragrant pear and its bud mutation materials. YB is Korla fragrant pear bud mutation material and CK is Korla fragrant pear. Figure (**A**) is the result of FFSR and CAR index values. Figure (**B**) is the result of AFW and ER index values. Figure (**C**) is the result of FDR, FSH, FPH and SI index values. Figure (**D**) is the result of SSO, SSU, SAR and WC index values. Figure (**E**) is the result diagram of TA, SC, VC and SP index values. FFSR: flower fruit set rate; CAR: calyx−abscission rate; AFW: average single fruit weight; ER: edible rate; FRD: fruit relative density; FSH: fruit skin hardness; FPH: fruit peel hardness; SI: shape index; SSO: soluble solids; SSU: soluble sugar; SAR: sugar:acid ratio; SC: stone cell; VC: vitamin C; SP: soluble protein. Two stars indicate that there was a significant difference at $p < 0.01$, and ns indicates no difference. The data were represented as mean $\pm$ SD with at least three biological repeats. Student's $t$−test was utilized for statistical analysis.

### 3.2. Korla Fragrant Pear and Its Bud Mutation Material Sepal (Cylinder) Shedding Cell Morphology Structure

Microscopic observation of cell morphology at the calyx (tube) junction (Figure 5) showed that, in the YB group, the abscission-layer cells at the junction of the calyx tube and young fruit were observed in the first period. The size of the abscission-layer cells was uniform and significantly smaller than that of the surrounding cells. In the second period, the detachment cells were observed to break. More abscission-layer cells were observed in the third and fourth sections, and abscission-layer cells were observed at the junction of the sepals and calyx tubes. In the CK group, no obvious detachment cells appeared in the first period, and during the second period, a few abscission-layer cells began to appear at the junction of the sepal and the calyx tube. In the third period, numerous abscission cells were observed at the junction of the sepals and young fruits. The results of this research demonstrate that the abscission-layer cells at the junction of the calyx tube and young fruit of Korla fragrant pear bud mutant material appeared earlier than those of Korla fragrant pear, and the bud mutant material of Korla fragrant pear formed the calyx fruit earlier.

### 3.3. GBS Detection of Korla Fragrant Pear and Its Bud Mutation Material

The high-quality SNP loci obtained are shown in Table 1. The total number of SNPs in 20 Korla fragrant pear materials was 713,605, and the number after filtration was 114,911. Figure 6 shows the phylogenetic tree obtained after 1000 times of calculations of the guide value, where the length of the evolutionary branch (genetic variation) was short, the length was similar, and the degree of variation was small. The bud mutation material of Korla fragrant pears was closely related to that of Korla fragrant pears. The principal component analysis results (Figure 7) demonstrate that the 17 samples were dense, indicating that the genetic differences between Korla fragrant pear bud mutation materials and Korla fragrant pears were small.

**Table 1.** Statistical results of SNP loci of 20 Korla fragrant pear materials.

| Genome of Reference | Number of Samples | Filter Condition | Total Number of SNP | Total Number of SNP after Filtering |
| --- | --- | --- | --- | --- |
| Pyrus bretschneideri | 20 | Dp4-miss 0.4-maf 0.01 | 713,605 | 114,911 |

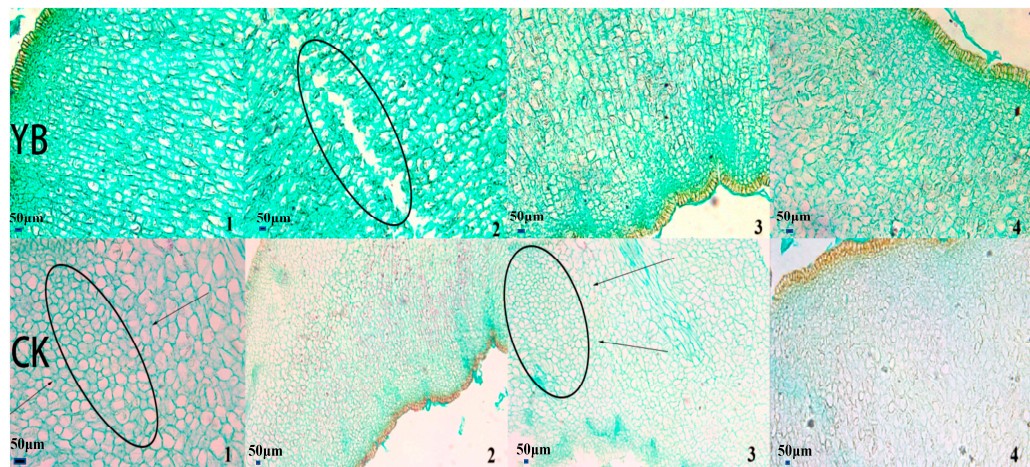

**Figure 5.** Paraffin section microphotograph of calyx off-junction tissue. YB is Korla fragrant pear bud mutation material, and CK is Korla fragrant pear. YB3, YB4, CK2, and CK3 were sepal shedding, and YB1, YB2, CK1, and CK4 were calyx shedding. YB1, YB2, YB3, and YB4 are pictured under a 5× objective lens; CK1 is pictured under a 10× objective lens; and CK2, CK3, and CK4 are pictured under a 2.5× objective lens. The first period was 2 h, 4 h, 6 h, 8 h, and 10 h after calyx tube discoloration. The second period was 12 h, 24 h, and 36 h after calyx tube discoloration. The third period was 48 h, 72 h, and 96 h after calyx tube discoloration. The fourth period was 120 h, 144 h, and 168 h after calyx tube discoloration.

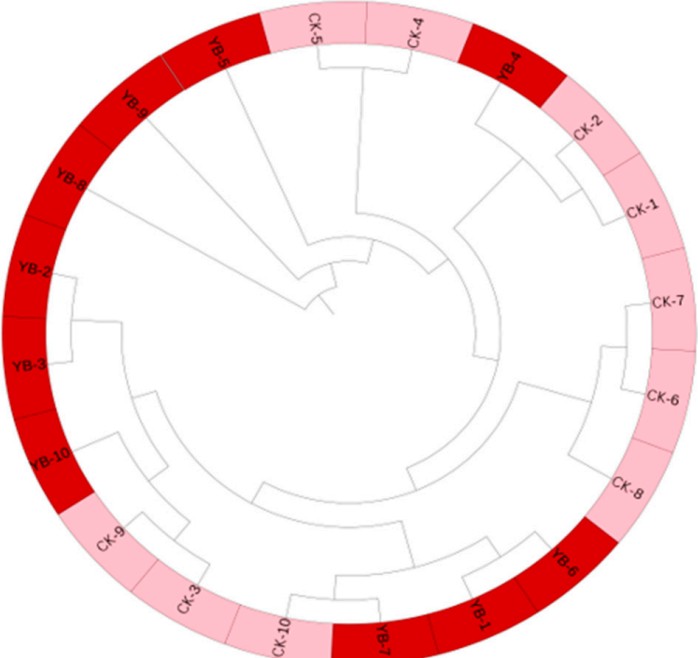

**Figure 6.** A phylogenetic tree was constructed using 114,911 SNPs from 20 Korla fragrant pear materials by the neighbor-joining method. The guide value is set to 1000 times. Different colors represent different germplasm materials. Dark red indicates 10 samples of bud mutation materials (YB1–YB10), and light red indicates 10 samples of Korla fragrant pear materials (CK1–CK10).

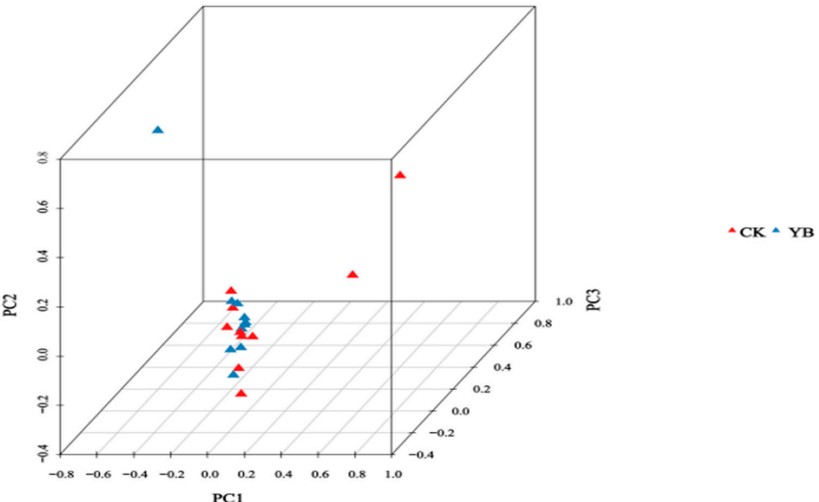

**Figure 7.** Principal component analysis of 20 Korla fragrant pear materials based on 114,911 SNPs. The coordinates in the figure represent principal component 1, principal component 2, and principal component 3, respectively. There are 3 error samples. The blue triangle represents 10 samples of mutant materials, and the red triangle represents 10 samples of Korla fragrant pear.

Figure 8 depicts the cross-validation error (CV error) extracted when K = 2–6 for the population genetic structure analysis of the Korla fragrant pear and its bud mutation materials. From the genetic structure diagram, with the decrease in the K value, the branches of the 20 materials became increasingly smaller. When K = 2, they were clustered into two branches. In addition, YB and CK were clustered together, indicating that they could not be divided into two subgroups.

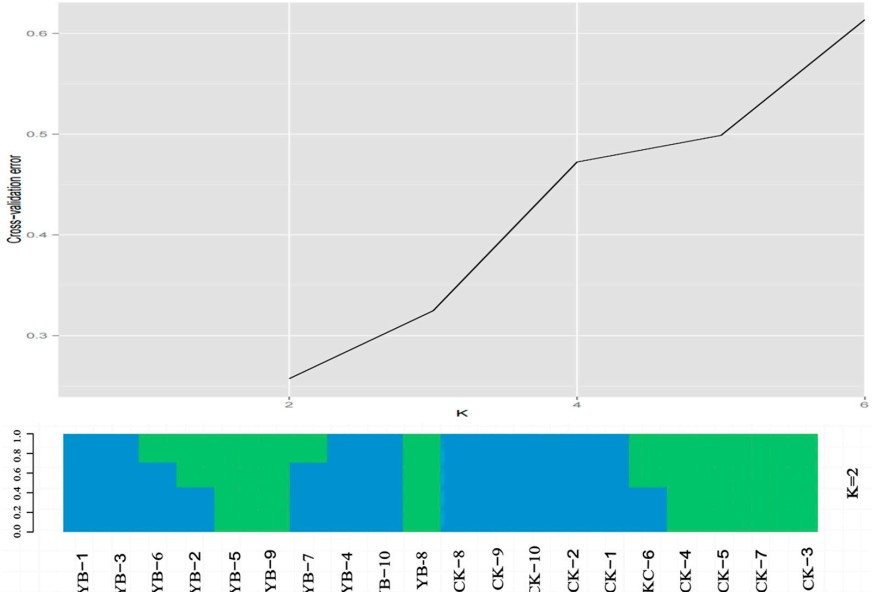

**Figure 8.** Population genetic structure analysis of 20 Korla fragrant pear materials based on 114,911 SNPs. The figure above is a line chart of the cross-validation error rate when the K value ranges from 2 to 6. The abscissa is the group subgroup number K value, and the ordinate is the cross-validation error rate. The following figure is the corresponding genetic structure distribution map when the K value is 2. Each square column represents one sample. YB1–YB10 represents 10 samples of bud mutation materials. PT1–PT10 represents 10 samples of Korla fragrant pear. The color of the square column represents the subgroup to which the sample belongs. The color ratio of the square column represents the proportion of the ancestral source of the sample.

The heterozygosity and genetic diversity indices of the Korla fragrant pear and its mutant materials were analyzed to reflect the degree of genetic variation in the population. Table 2 shows the results. The observed heterozygosity of YB was 0.7757, the expected heterozygosity was 0.4276, and the genetic diversity index was 0.4545. The observed heterozygosity of the CK group was 0.7726, the expected heterozygosity was 0.4285, and the genetic diversity index was 0.4560. The data show that there was no significant difference between the two sample groups YB and CK ($p > 0.05$). The genetic diversity between the two groups was similar, and the two groups could be classified as the same group.

**Table 2.** Level of population genetic diversity in the YB and CK groups. Different lowercase letters in the same column indicate significant differences ($p < 0.05$).

| Population (Variety) | Observed Heterozygosity (Ho) | Expected Heterozygosity (He) | Genetic Diversity Index (pi) |
|---|---|---|---|
| YB | 0.7757 a | 0.4276 a | 0.4545 a |
| CK | 0.7726 a | 0.4285 a | 0.4560 a |

*3.4. SSR Molecular Markers of Korla Fragrant Pear and Its Bud Mutation Materials*

A total of 5736 SSR markers were detected, and 5429 SSR primers were successfully designed. The success rate of the primer design was 94.67%. Among the 5429 primers, dinucleotides and trinucleotides were the main types (Figure 9). Moreover, there were 1535 dinucleotides and 1642 trinucleotides in the bud mutation materials, and 1554 dinucleotides and 1694 trinucleotides in Korla fragrant pear. Unique SSR primers of the YB and CK groups were screened, and 249 pairs of primers were identified. The detection results (Table S1) demonstrate that the OD260/OD280 values of the two DNA samples in the YB and CK groups were between 1.8 and 2.0 and that the extracted DNA sample solution was of high purity and quality and could be used for subsequent experiments.

In order to further verify the genetic background of the YB group and CK group, a total of 249 pairs of primers were used to amplify the DNA from the two parents. The polyacrylamide gel electrophoresis results (Figure 10) demonstrate that 233 pairs of primers in the 249 pairs of primers did not show different bands in the lane, and 16 pairs of primers showed different bands in the lane, indicating that the bud mutation material of Korla fragrant pear and Korla fragrant pear differed at the DNA level. These 16 pairs of primers can distinguish two varieties. Among them, 11 pairs of primers were synthesized using SSR-specific sequences of the YB group (primer numbers: 1, 2, 3, 5, 6, 7, 8, 10, 12, 14, and 16), and five pairs of primers were synthesized using SSR-specific sequences of the CK group (primer numbers: 4, 9, 11, 13, and 15). Table S2 shows details of the 16 primer pairs.

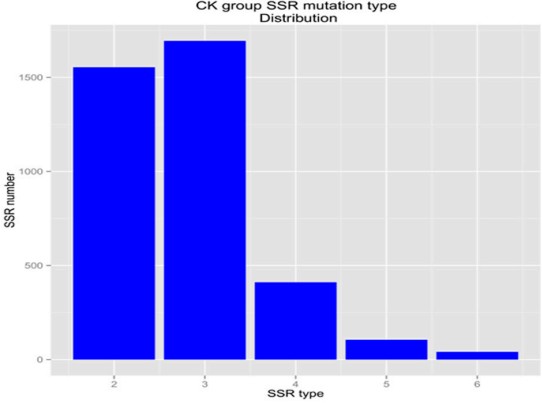

**Figure 9.** *Cont.*

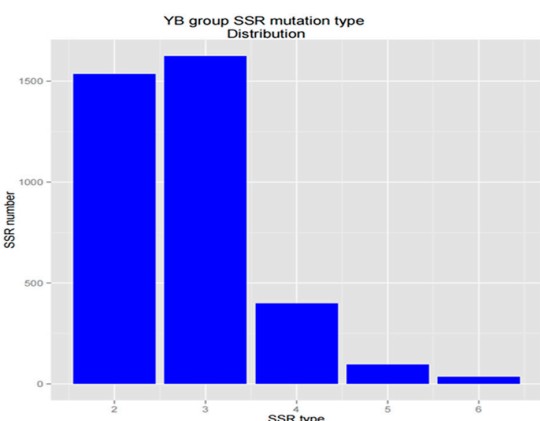

**Figure 9.** SSR primer type density distribution map of Korla fragrant pear and bud mutation materials. The YB in the lower image represents the bud mutation material, and the CK in the upper image represents the Korla fragrant pear. The abscissa in the figure represents the SSR primer type. Number 2 represents the dinucleotide type, and number 3 represents the trinucleotide type. The ordinates in the figure represent the number of SSRs corresponding to the primer type.

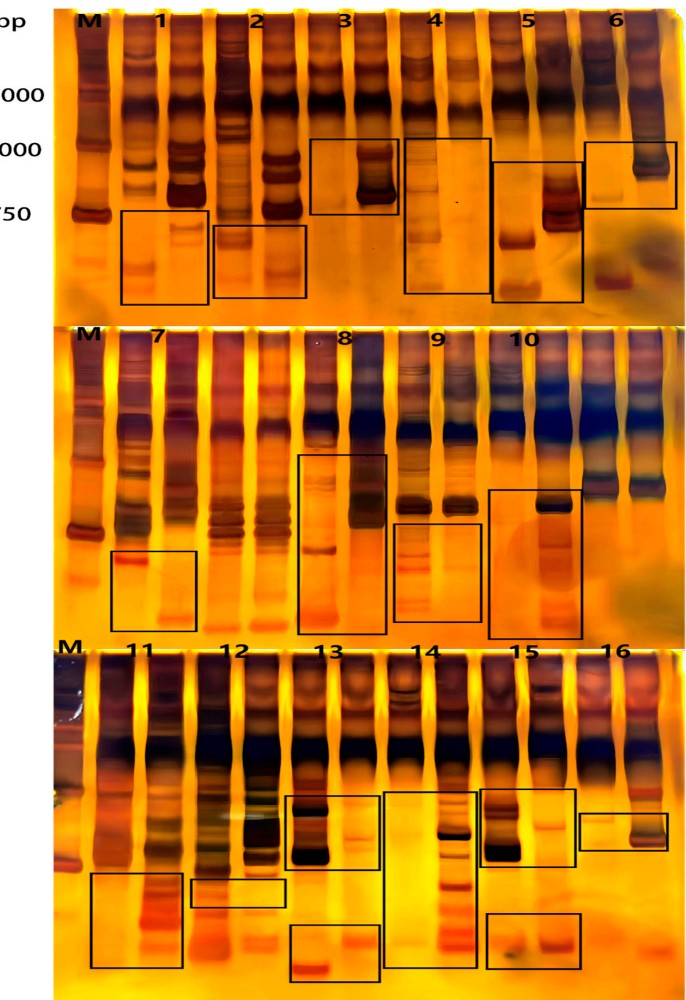

**Figure 10.** The results of polyacrylamide gel electrophoresis silver staining of mutant materials and Korla fragrant pear under SSR primer amplification. The first column on the left side of the diagram is the bp value corresponding to the maker strip. The numbers 1–16 indicate 16 pairs of primers. The black boxes in the figure indicate the difference between the bands amplified by 16 pairs of primers between the mutant material and the Korla fragrant pear.

## 4. Discussion

### 4.1. Fruiting Characteristics Analysis of Korla Fragrant Pear Bud Mutation Material

Based on numerous long-term investigations and studies by our team, we observed the bud mutation material of Korla fragrant pear with specific changes in seed-setting characteristics and confirmed by years of grafting and cultivation as a stable genetic variation. This bud mutation material is the same as the phenological period of Korla fragrant pear, and there is no significant difference between the two (Figures S1 and S2; Tables S3–S6). This bud mutation material has been approved by the Forest Variety Approval Committee of the Forestry and Grassland Bureau of Xinjiang Uygur Autonomous Region of China (improved variety number: New S-SV-PS-008-2023). The significant differences may exist in the seed-setting characteristics of the two.

Actually, there is self-incompatibility in the flowers of Korla fragrant pear during pollination [57]. Therefore, in the actual production process, artificial and liquid pollination are used to improve the fruit-setting rate of fragrant Korla pears. In this research, the natural pollination group of Korla fragrant pear bud mutation material and the natural pollination group of Korla fragrant pears were established. The fruit-setting rates of the flowers were investigated and compared. Under natural conditions, the fruit-setting rate of the natural pollination group of the Korla fragrant pear bud mutant material was significantly higher than that of the natural pollination group of the Korla fragrant pear. This may be related to the stigma receptivity of Korla fragrant pear bud mutation materials, and an increase in stigma receptivity or receptivity time may improve pollination and fruit setting [58]. The high fruit-setting rate of Korla fragrant pear bud mutation material may be related to its sensitivity to temperature changes. The flowering period of Korla fragrant pear was significantly negatively correlated with the average temperature in winter, the temperature in spring, and the soil temperature at different depths, and was significantly positively correlated with the date of spring and the date when the average temperature was over 10 °C for 5 consecutive days [59]. The bud mutation material of Korla fragrant pears may be more sensitive to temperature changes and can perceive external temperature changes earlier and make feedback adjustments. The high fruit-setting rate of Korla fragrant pear mutants may be due to the gene mutation of the multiple allele (*S-allele*) that controls style and pollen specificity. The determinant *S-Rnase*, which controls style self-incompatibility, and *SLF*, which controls pollen self-incompatibility, cannot function normally [60,61]. Whether the high fruit-setting rate of Korla fragrant pear bud mutation material is caused by stigma receptivity, temperature change, and multiple alleles (*S-allele*) alone or in combination remains to be further studied.

Whether the calyx falls off during fragrant Korla pear growth is an important physiological process that affects their quality. The stone cell cluster content in the fruit with persistent calyces was much higher than that in the fruit with calyx shedding, and the sugar content was lower than that in the fruit without calyces, which seriously affected the quality of Korla fragrant pears. The results of this research demonstrate that the calyx-removal rate of Korla fragrant pear bud mutants under natural conditions was significantly higher than that of Korla fragrant pear. Studies on sepal abscission have shown that sepal abscission is related to hormones, miRNAs, and certain enzymes. For example, studies have reported that sepal abscission is related to fruit position and the content of GA3, IAA, and ABA in the calyx end during calyx abscission [62,63]. Studies have reported that miRNAs differentially expressed between sepal abscission and persistence are involved in regulating various biological processes [7,64,65]. These processes are primarily related to steroid biosynthesis, starch and sucrose metabolism, and galactose metabolism. Target genes of the nine miRNAs were related to calyx persistence in fragrant Korla pears. Studies have reported that polygalacturonases and chitinases play key roles in calyx shedding [66]. Whether the high calyx-removal rate of Korla fragrant pear bud mutation material is caused by hormones, miRNA, and enzymes alone or in combination requires further research.

The results of this experiment demonstrate that there was no significant difference in appearance or internal quality between Korla fragrant pear mutants and Korla fragrant

pear, and the fruit quality of Korla fragrant pear mutants was not affected. Compared to Korla fragrant pears, existing bud mutation varieties (lines) have improved in terms of single fruit weight and maturity, but they lead to a decrease in certain quality index values. For example, the soluble sugar content and starch content of 'Xinli 2' decreased, and the soluble solids content and soluble sugar content of 'Sha 01' decreased [19,20]. Quality indicators, such as sugar, acid, soluble solids, and stone cells, affect the Korla pear flavor [67]. Changes in one or more of these indicators affect the Korla pear flavor. The Korla fragrant pear mutants studied in this experiment showed great advantages in terms of the calyx-removal rate and fruit-setting rate but did not affect fruit quality and better retained the original flavor of Korla fragrant pear. Therefore, as a new germplasm resource for Korla fragrant pears, the bud mutation material of Korla fragrant pears should be better developed and utilized in the future.

*4.2. Anatomical Structure Analysis of Sepals and Calyx Tube of Korla Fragrant Pear Bud Mutation Material*

Paraffin section technology can be used to research changes in tissue structure in different plant organs, and the developmental characteristics of tissue structures can be mastered [10]. However, there are few studies on the anatomical structure of the calyx and calyx tube of Korla fragrant pears and whether calyx removal seriously affects the fruit quality of Korla fragrant pears. Therefore, this research investigated the bud mutation materials of Korla fragrant pear and the sepal (cylinder) tissue of Korla fragrant pear to understand the characteristics of its tissue structure changes to provide a theoretical anatomical basis for sepal shedding. Abscission of Korla fragrant pear sepals occurred after abscission zone formation. The abscission zone is the area where plant organs are connected to the plant matrix, and its cells are small and dense. The abscission layer is the area where cell separation occurs in the abscission zone [68,69]. Several studies have been conducted on detached cells. Anatomical studies were conducted on the longitudinal sections of sorghum florets during different periods. The results demonstrated that in sorghum, an abscission zone formed in the pedicel below the spikelet, composed of several to multiple cell layers, and these cells were significantly smaller than the surrounding adjacent cells [70]. Studies have reported the anatomical structure of petiole abscissions in *Arabidopsis thaliana*. The filament-, petal-, and sepal-receptacle boundaries of wild-type Arabidopsis stamens show closely arranged abscission cells [71]. This research observed that abscission-layer cells could be observed within 2, 4, 6, 8, and 10 h after calyx tube discoloration of Korla fragrant pear bud mutant material, and the size of the abscission-layer cells was uniform and significantly smaller than that of the surrounding cells. The results of this research are consistent with those of previous studies. This research observed that the abscission-layer cells of the Korla fragrant pear bud mutation material appeared earlier than those of the Korla fragrant pear, which may be related to the type and content of hormones and enzymes in the bud mutation material and may be related to the key genes controlling abscission-layer cell development. For example, studies have demonstrated that *JOINTLESS*, a family gene of *MADS*, can control abscission zone development and that *JOINTLESS2* can regulate the proliferation of abscission-layer cells [72]. Changes in hormone content and gene expression may accelerate the development of the calyx area in the bud mutant material and form abscission-layer cells. The reason why the abscission-layer cells of Korla fragrant pear bud mutant material appeared earlier than those of Korla fragrant pear requires further research.

*4.3. Genetic Structure Analysis of Korla Fragrant Pear Bud Mutation Materials*

Population genetic diversity analysis is an effective method to research the genetic background of different germplasm resources [73]. GBS genotyping by sequencing is a low-cost, high-throughput genotyping method. GBS technology avoids the shortcomings of traditional genotyping methods and has been widely used in genome-wide association analyses, gene diversity research, and germplasm identification [74]. In the research of the

genetic diversity of 96 olive (*Olea europaea*) germplasm resources [75] in the database of the United States Department of Agriculture, it was found that 96 germplasm resources were divided into 7 subpopulations based on GBS sequencing. These seven subpopulations are inconsistent with geographical origin, indicating that they may be caused by regional selection. In the research of genetic diversity of 219 germplasm resources of *Phaseolus vulgaris* [76], it was found that 219 germplasm resources were divided into two gene pools (Mesoamerican gene pool and Andean gene pool). At the same time, it was found that there were several unique SNPs in the two gene pools. In addition, GBS technology was also used to analyze 150 jujube (*Ziziphus jujuba*) germplasms [77]. A total of 4680 SNPs were found, and 150 germplasms were divided into two groups. In our results, it is interesting to note that Korla fragrant pears and budding materials are not divided into two categories by GBS sequencing. This is slightly different from previous research results. We speculate that although the bud mutation materials are obviously expressed in the apparent traits, the degree of variation in the genetic diversity of the population is small, which is not enough to achieve the degree of population evolution, so there is no group differentiation. At the same time, because there is no reference genome for Korla fragrant pear, we selected white pear as the reference genome for comparison. There may be differences between the genomes, so that some SNP loci unique to Korla fragrant pear may not be developed.

At present, molecular marker technology has been widely used in plant genetic breeding. Among them, SSR molecular marker technology has certain advantages in variety identification and genetic relationship analysis [78]. In the research of the identification of 423 cabbage (*Brassica campestris* ssp. *chinensis*) varieties [79], 23 markers were finally found by SSR molecular marker technology. These 23 markers can distinguish 418 varieties. In the research of genetic diversity of 97 chrysanthemum (*Chrysanthemum morifolium*) varieties [80], it was found that 97 varieties were successfully divided into three categories by 14 SSR markers. The three categories are small flower varieties, medium flower varieties, and large flower varieties. In addition, 19 SSR markers were used to distinguish 121 tomato (*Solanum lycopersicum*) germplasm resources [81]. Interestingly, based on the SSR molecular marker method, we successfully distinguished Korla fragrant pear and its mutant materials by 16 pairs of primers after screening. This is similar to the results of previous studies. We speculate that although there is no group differentiation at the GBS level, the two are indeed different at the DNA level. This also confirms that SSR molecular marker technology can be used to distinguish varieties with a close genetic relationship. In addition, we speculated that the gene fragments corresponding to these 16 pairs of primer sequences may be related to the bud mutation traits and may be able to distinguish the bud mutation materials from Korla fragrant pear. However, functional verification needs to be carried out in the future in order to develop and utilize these 16 pairs of primers more accurately.

## 5. Conclusions

Based on the investigation of seed-setting characteristics, observation of anatomical structure, and molecular marker methods, this research finally confirms that the mutant material we found was a good germplasm resource. Our results enrich the germplasm resources of Korla fragrant pear breeding. The bud mutation material can be used as a new variety of simplified cultivation of Korla fragrant pear in Bayinguoleng Mongolian Autonomous Prefecture of Xinjiang, China.

In short, we have bred new varieties suitable for local planting for the Korla fragrant pear market. This is of great significance in promoting the optimization of the Korla fragrant pear variety structure and the healthy development of the industry. At the same time, it lays a theoretical foundation for exploring related trait genes through genome-wide association analysis (GWAS) in the future.

**Supplementary Materials:** The following supporting information can be downloaded at: https://www.mdpi.com/article/10.3390/app14156589/s1.

**Author Contributions:** Conceptualization, C.Z. and S.W.; methodology, Y.C.; software, X.Y.; validation, C.Z., S.W. and X.Y.; formal analysis, X.Y.; investigation, L.Z.; resources, H.S.; data curation, X.Y.; writing—original draft preparation, X.Y.; writing—review and editing, S.W.; visualization, C.Z.; supervision, S.W.; project administration, C.Z.; funding acquisition, S.W. All authors have read and agreed to the published version of the manuscript.

**Funding:** This research was supported by the 2022 Central Finance Forestry and Grass Science and Technology Extension Demonstration Project 'Demonstration and Extension of Standardized and Efficient Cultivation Techniques of Korla Fragrant Pear' (New [2022] TG20).

**Institutional Review Board Statement:** Not applicable.

**Informed Consent Statement:** Not applicable.

**Data Availability Statement:** The data presented in this study are available in.

**Acknowledgments:** We sincerely thank everyone who was involved in this research and Supplementary Material.

**Conflicts of Interest:** The authors declare no conflicts of interest.

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
