# Peer review of "Fruiting Characteristics and Molecular-Assisted Identification of Korla Fragrant Pear Bud Mutation Materials"

_applsci, doi:10.3390/app14156589_

Round 1

Reviewer 1 Report

Comments and Suggestions for Authors

The authors develop an interesting topic for the study of the germplasm available for a species native to China, but which is currently making its way towards new international markets. However, certain aspects in the manuscript need to be improved to make it more understandable.

The abstract must include all the variables that were evaluated both in the morphological and molecular parts. Finish this item with the impact of the results found.

In the introduction, background studies must consider the international or regional context, the species, and the main conclusions. Make clear the justification for the work.

In the results there are some figures whose resolution is not very good. Likewise, improve the legends of the tables and figures, since they should be more explanatory. In the molecular part, it is recommended to eliminate table 3 since the information it presents is not very relevant to the study, rather select another table that can strengthen more results.

The discussion contains important biological aspects that effectively support the results found, however, it is recommended to complement the discussion from a genetic and agronomic point of view, showing their practical impact.

In the conclusion item, it is suggested to summarize it and mainly show the impact of the study results. Use more up-to-date references.

Review other revisions in the manuscript.

Reviewer 2 Report

Comments and Suggestions for Authors

Dear authors,

Hope you are doing well.

I have read and reviewed your manuscript in detail and believe that it needs to be improved in certain aspects that I indicate below. Of course, I think that the experimental design is correct and the concept of the paper is original.

Major concerns

1.       Introduction. I think the introduction is a bit long. I consider that some aspects could be omitted in this section and included as part of the discussion. For example, Paraffin section technology, GBS genotyping,... I think it is not necessary to describe advantages and disadvantages of these technologies, since the structural or genetic aspects themselves are more important than these.

2.       Line 118. “(Figure S1-S2; TableS1-S4;).” From my point of view, this information should not be included in this section. I consider that it is more correct include it as part of (results) or/and discussion section. Additionally, if information on phenology or other data has been obtained from other authors, they must be cited in the legend of the figure or table.

3.       Materials and methods. “GBS gene library construction and identification and SNP detection should include more information about the procedure.

4.       Materials and methods. Statistical Analysis.  It is essential that this section includes information about p-values, biological replicates and other relevant information.

5.       Part of the information generated in the research appears in the results and is not mentioned in the material and methods section. Furthermore, the results include a lot of information that should be included in materials and methods.

6.       Discussion. Lines 513-533. This part of the discussion seems a results description. Therefore, they must be improved, highlighting the main findings and discussing them with the corresponding bibliographic references.

7.       Conclusions. This section should skip study objectives and focus on the main findings and possible biological implications and their applications. Sometimes it seems like a repeat of the results. Remember that you are publishing your results in an applied science journal.

8.       As a general tip, you should include a brief final paragraph in the discussion or conclusions section to demonstrate the applied potential of this research. When wanting to publish in an applied science journal, therefore, with a high practical component, readers must be able to identify its applications. This comment is totally personal, and I would understand that these findings could be subject to legal protection.

Minor concerns

1.       From my opinion p-values should be removed from the abstract, introduction and discussion. These details can be found by the authors in the results section.

2.       Line 67. “(m13144)” could be removed.

3.       Line 118. “And this bud mutation material”. This could be replaced with something like “Besides, this bus mutation material….”.

4.        Line 153. Cardinal points should be written with capital letter (North,…).

5.       After ":" no capital letter is written.

6.       Line 179. “transverse diameter and longitudinal diameter”. Please, i.ndicate the units of measurement. The same for the rest of parameters.

7.       Figures 3 and 4. Please, include scale bar in photos obtained from microscope.

8.       Description of Figure 4. I think that it is necessary to clarify to readers the meaning of “first period”, “second period”…

9.       “Table 1. Statistics of SNP.” The name of this table can be uninformative for redaders.

10.  Figure 5. Please, include more details about the phylogenetic tree in the legend and bootstrap values in the graph.

11.  Figure 6, 7, 8 and 9. Please, include more information in the legend.

12.  Line 463. Could you change “The results of this study” by “The results of this experiment”?

13.  Line 489. Arabidopsis thaliana must be written in italics.

14.  Gene names must be written in italics. For Instance, lines 500, 501.

15.  Line 535. I suggest “In this research,…”.

Best regards,

Round 2

Reviewer 2 Report

Comments and Suggestions for Authors

Dear authors,

Thank you for considering my previous comments. Also congratulations for the work.

Best regards,